# Growth hormone-releasing hormone disruption extends lifespan and regulates response to caloric restriction in mice

Liou Y Sun[1][†]*, Adam Spong[1][†], William R Swindell[2], Yimin Fang[1], Cristal Hill[1], Joshua A Huber[1], Jacob D Boehm[1], Reyhan Westbrook[1], Roberto Salvatori[3], Andrzej Bartke[1]

[1]Department of Internal Medicine, Southern Illinois University School of Medicine, Springfield, United States; [2]Department of Dermatology, University of Michigan School of Medicine, Ann Arbor, United States; [3]Division of Endocrinology and Department of Medicine, Johns Hopkins University School of Medicine, Baltimore, United States

**Abstract** We examine the impact of targeted disruption of growth hormone-releasing hormone (GHRH) in mice on longevity and the putative mechanisms of delayed aging. GHRH knockout mice are remarkably long-lived, exhibiting major shifts in the expression of genes related to xenobiotic detoxification, stress resistance, and insulin signaling. These mutant mice also have increased adiponectin levels and alterations in glucose homeostasis consistent with the removal of the counter-insulin effects of growth hormone. While these effects overlap with those of caloric restriction, we show that the effects of caloric restriction (CR) and the GHRH mutation are additive, with lifespan of GHRH-KO mutants further increased by CR. We conclude that GHRH-KO mice feature perturbations in a network of signaling pathways related to stress resistance, metabolic control and inflammation, and therefore provide a new model that can be used to explore links between GHRH repression, downregulation of the somatotropic axis, and extended longevity.

*For correspondence: leeosun@gmail.com

[†]These authors contributed equally to this work

**Competing interests:** The authors declare that no competing interests exist.

**Reviewing editor**: David M Sabatini, Whitehead Institute and the Massachusetts Institute of Technology, United States

## Introduction

Genetic studies in a variety of organisms have revealed that the endocrine systems play a central role in lifespan determination (*Tatar et al., 2003*). For example, mutations in the insulin/Insulin-like growth factor I (IGF-I) signaling can robustly increase longevity in the nematode *C. elegans* (*Kenyon, 2010*), whereas in mouse, disruptions in growth hormone (GH) pathway dramatically prolong lifespan (*Bartke, 2011*).

Ames and Snell dwarf mice are the most studied mutants in which altered GH signals produce dramatic increases in lifespan (*Brown-Borg et al., 1996*; *Flurkey et al., 2001*). A number of aging-related phenotypes are also delayed in these mice, including collagen cross-linking, cataract development, kidney diseases, fatal neoplastic diseases and decline in immune function, locomotor activity, learning and memory (*Bartke, 2011*). In these models, homozygous mutation of either Prop1 or Pit1 genes cause an abnormal development of the anterior pituitary, which in turn leads to decline in production of GH, thyrotropin (TSH), and prolactin (PRL), with consequent decrease in circulating IGF-I and thyroxine levels (*Bartke, 2011*). The specific contribution of GH signaling to lifespan extension in these systems is supported by studies of downstream pathway elements. For instance, mice with disruption of the GH receptor (Ghr−/−) have also markedly increased lifespan with concomitant delay of late life diseases and disabilities (*Coschigano et al., 2003*; *List et al., 2011*). These findings support the hypothesis that dampening of the GH pathway is the key contributor to lifespan

**eLife digest** There is increasing evidence that the hormonal systems involved in growth, the metabolism of glucose, and the processes that balance energy intake and expenditure might also be involved in the aging process. In rodents, mutations in genes involved in these hormone-signaling pathways can substantially increase lifespan, as can a diet that is low in calories but which avoids malnutrition. As well as living longer, such mice also show reductions in age-related conditions such as diabetes, memory loss and cancer.

Many of these effects appear to involve the actions of growth hormone. Mice with mutations that disrupt the development of the pituitary gland, which produces growth hormone, show increased longevity, as do mice that lack the receptor for growth hormone. However, these animals also show changes in a number of other hormones, making it difficult to be sure that the reduction in growth hormone signaling is responsible for their increased lifespan.

Now, Sun et al. have studied mutant mice that lack a gene called GHRH, which promotes the release of growth hormone. These mice, which have normal levels of all other pituitary hormones, lived for up to 50% longer than their wild-type littermates. They were more active than normal mice and had more body fat, and showed greatly increased sensitivity to insulin.

Some of the changes in these mutant mice resembled those seen in animals with a restricted calorie intake, suggesting that the same mechanisms may be implicated in both. However, Sun et al. found that caloric restriction further increased the lifespans of their GHRH knockout mice, indicating that at least some of the effects of caloric restriction are independent of disrupted growth hormone signaling.

The results of this study are an important step forward for understanding how growth hormone signaling and caloric restriction regulate aging, both individually and in combination. The GHRH knockout mice are likely to become an important model system for studying these processes and for understanding the complex interactions between diet and hormonal pathways.

extension in mice. Nevertheless, the associated lack of TSH and prolactin makes these two models less than optimal in conclusively exclude the influence of the lack of these hormones on the delayed aging phenotype.

Caloric restriction (CR) has been shown to extend lifespan in many species and has been extensively used in experimental gerontology to modulate development of age-related diseases (*Weindruch and Sohal, 1997*). In rodents, CR delays the onset of cancer, atherosclerosis, and diabetes, and typically increases lifespan (by 15% in mice and by 30% in rats) (*Swindell, 2012*). Although this phenomenon was first described over 70 years ago, the molecular basis mediating the effects of CR on the aging process remains incompletely understood. Intriguingly, phenotypic characteristics of the long-lived mutant mice with disrupted GH axis overlap with some effects of CR, suggesting possible mechanistic connections. Shared characteristics include: (a) small body size; (b) reduced blood glucose and increased insulin sensitivity; and (c) reduced or absent levels of various hormones and growth factors, that is, GH, insulin, and IGF-I; (d) delaying and/or suppression of the occurrence of several age-related diseases. Nevertheless, longevity phenotypes in different mouse models may rely on CR-sensitive pathways to varying degrees. For instance, 30% CR confers additional life extension in Ames dwarf mice (*Bartke, 2011*), but has no additional effect on longevity in male Ghr−/− mice, and only a modest reduction of late-life mortality in Ghr−/− females (*Bonkowski et al., 2006*).

In this study, we examined the longevity of mice with isolated GH deficiency due to targeted disruption of the GHRH gene (GHRH-KO). This gene is required for somatotroph cell proliferation and GH secretion (*Alba and Salvatori, 2004*). We provide a phenotypic, metabolic and molecular-level characterization of GHRH-KO mice and show that GHRH-KO mutants exhibit lifespan extension comparable to the Ames and Snell dwarf mice. Moreover, we have shown that, in contrast with the Ghr−/− mice, lifespan in GHRH-KO mice is further extended by CR. These findings established the GHRH-KO mice as a novel rodent model for delayed aging and implicate CR-independent mechanisms in longevity assurance.

## Results

### Robustly increased longevity in GHRH-KO mice

To investigate the effect of isolated GH deficiency on lifespan, we evaluated differences in longevity of GHRH-KO (KO) mice and littermate (wild-type) control mice on ad libitum (AL) standard diet. As shown in *Figure 1A*, median survival of GHRH-KO mice (sexes combined) was increased by 295 days (or 46%) relative to that of control mice (931 days for KO mice vs 636 days for control mice). This difference in survival between KO mice and controls was significant based upon a non-parametric test (p<0.001; log-rank test).

Analysis of each sex separately showed that median survival in female GHRH-KO mice was increased by 290 days (43%; from 666 to 956 days) relative to that of control mice (logrank test, χ2 = 46.7, p<0.0001) (*Figure 1B,C*). Male GHRH-KO mice had increased median lifespan by 314 days (from 614 to 928 days) or 51% relative to that of control mice (logrank test, χ2 = 28.4, p<0.0001) (*Figure 1B,D*).

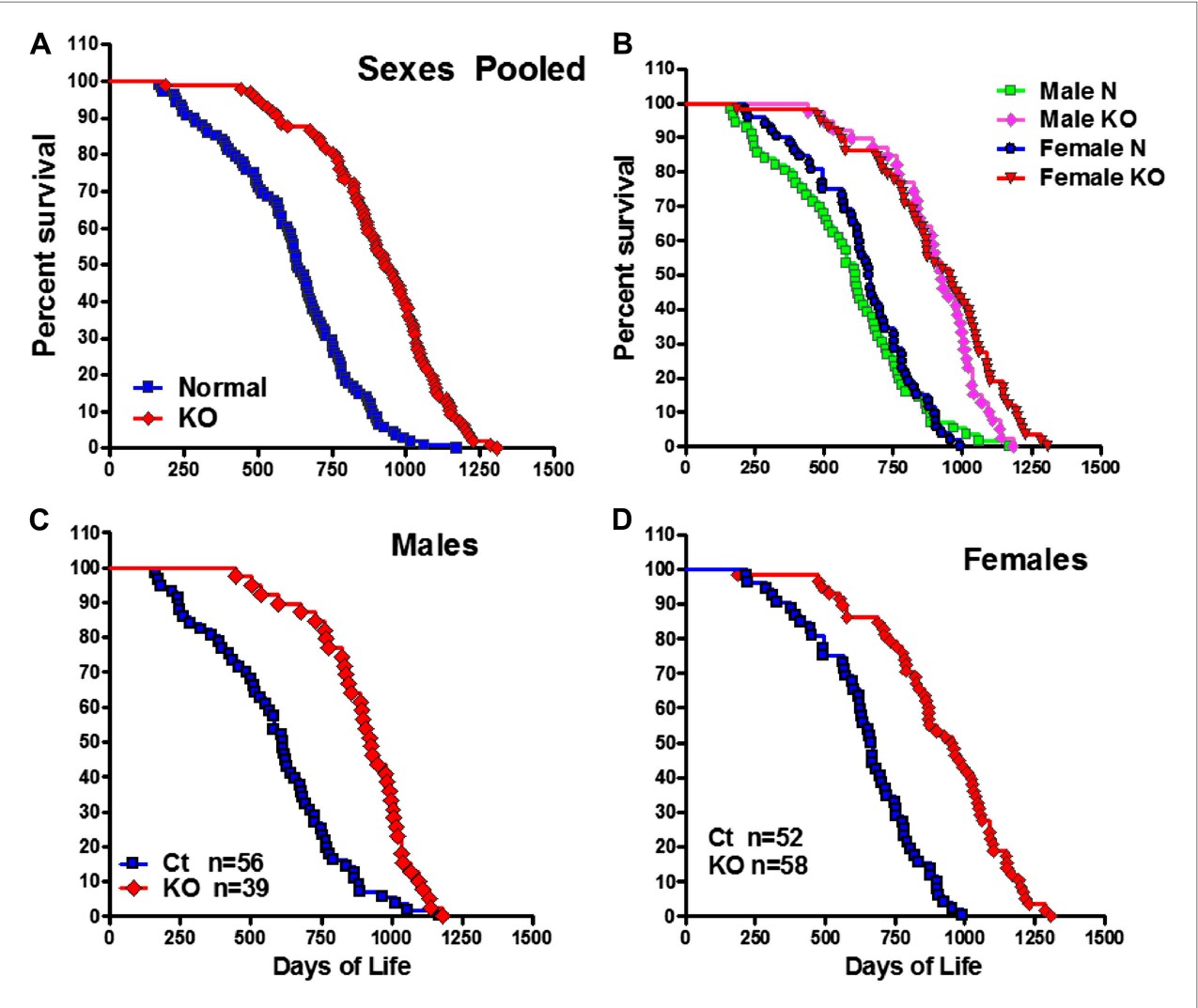

**Figure 1**. Increased longevity of GHRH-KO mice. Kaplan-Meier survival curves for each genotype: GHRH-KO (KO) and Control (Ct) mice; each point represents a single mouse. (**A**) Sex pooled survival curves. (**B**) Sex separated survival curves. (**C**) Male survival curves (N = 56 for controls, N = 39 for KO). (**D**) Female survival curves (N = 52 for controls, N = 58 for KO).

The following figure supplements are available for figure 1:

**Figure supplement 1**. Changes in body length of GHRH-KO mice (Red) and WT control mice (Black) from 1 to 8 weeks of age.

To evaluate the change of maximal lifespan, we used the Wang/Allison (*Wang et al., 2004*) method to compare the proportion of live mice in each group at the age at which only 10% of the population remained alive. GHRH deletion also led to a significant increase in maximum lifespan by 33% for females (p<0.05) and by 18% for males (p<0.05) relative to controls. The robust increase in both median and maximal lifespan in GHRH-KO mice is consistent with the notion that inhibition of GH signaling ameliorates age-related disease, potentially slowing the aging process.

## Phenotypic characteristics and blood parameters of GHRH-KO mice

Both male and female GHRH-KO (KO) mice were markedly smaller than normal littermate controls (*Figure 2A*). Although KO mice weighed the same as control animals at birth (data not shown), they gained less weight than control mice (*Figure 2A*). There was a significant reduction in body length in both male and female KO mice (*Figure 1—figure supplement 1*). Food consumption on a per gram body weight basis was not significantly different between KO and control mice. However body composition studies showed increased adiposity in both male and female KO mice (*Figure 2C*).

To shed light on the physiological mechanisms involved in the energy homeostasis of KO mice, we assessed the effect of isolated GH deficiency on energy expenditure (EE) by indirect calorimetry. To estimate fuel utilization, we used the respiratory quotient (RQ), which is a dimensionless ratio comparing the volume of carbon dioxide an organism produces to the volume of oxygen consumed over a given time (RQ = VCO2/VO2) (*Johnston et al., 2006*). The RQ varies inversely with lipid oxidation. A higher fasting RQ, which indicates lowered fat oxidation, is linked to body weight gain, metabolic inflexibility and insulin resistance (*Zurlo et al., 1990*). The KO mice had a significantly decreased RQ compared to their wild-type controls during both the dark and light periods under normal housing conditions (*Figure 2E*), although their oxygen consumption per gram of body weight (measured by VO2) was not different from controls (*Figure 2B*). Interestingly, on measures of spontaneous home cage activity (*Figure 2D*), KO mice were significantly more active than their wild-type counterparts.

Heightened insulin sensitivity is a common characteristic of several types of mice with GH-related mutations and extended longevity (*Bartke, 2011*). Consistent with this observation, plasma glucose concentrations were significantly reduced in both male and female GHRH-KO mice under fasted conditions (*Figure 2G*). Furthermore, the lower score of homeostasis model-assessment of insulin resistance (HOMA-IR) indicated that GHRH-KO mice had enhanced insulin sensitivity (*Figure 2G*). As expected from deficiency of pituitary GH, IGF-I levels in the circulation were much lower in GHRH-KO mice than controls. Moreover, insulin tolerance tests (ITT) done in mice fasted for 4 hr showed that plasma glucose concentrations decreased more in KO mice than in control mice (*Figure 2H*) despite of normal intraperitoneal glucose tolerance tests (IPGTT) results (*Figure 2F*). Together, these data indicate that isolated GH deficiency results in improved insulin sensitivity and glucose homeostasis.

Decreased ribosomal protein S6 kinase 1 (S6K1) activity has been shown to increase insulin sensitivity (*Um et al., 2006*). Since S6K1 signaling is decreased in the tissues of Ames dwarf mice (*Sharp and Bartke, 2005*), we speculated that S6K1 signaling may contribute to regulate insulin sensitivity in response to isolated GH deficiency. To investigate this possibility, we examined the levels of phosphorylation of ribosomal protein S6 and phosphorylation of IRS1 (*Figure 2I*), two downstream targets of S6K1 (*Um et al., 2006*). Hepatic levels of phosphorylation of these proteins were significantly lower in KO mice when compared with controls. Consistent with the evidence in S6k1−/− mice (*Selman et al., 2009*), these data suggest that lower activity of S6K1 in GHRH-KO mice might contribute to the improved insulin sensitivity.

## Hepatic gene expression profiles in GHRH-KO mice: partial overlap with CR and evidence for Nrf2 activation

Microarray analysis was performed to identify genes altered in liver tissue from GHRH-KO mice as compared to normal littermates (n = 3 per genotype). Overall, we identified 141 genes with elevated expression in the mutants (FDR < 0.05 and FC > 1.50; *Supplementary file 1A*), along with 164 genes with decreased expression (FDR < 0.05 and FC < 0.67; *Supplementary file 1B*). The most highly increased genes were *Sult2a2*, *Sult1e1* and *Spink3* (FC > 38; *Supplementary file 1A*), while the most strongly decreased genes were *Hsd3b5*, *Slco1a1* and *Igf1* (FC <0.02; *Supplementary file 1B*). The 141 increased genes were disproportionately associated with GO BP terms related to differentiation, development and proliferation, such as positive regulation of mononuclear cell proliferation (GO:0032946), regulation of multicellular organismal development (GO:2000026) and regulation of cell differentiation

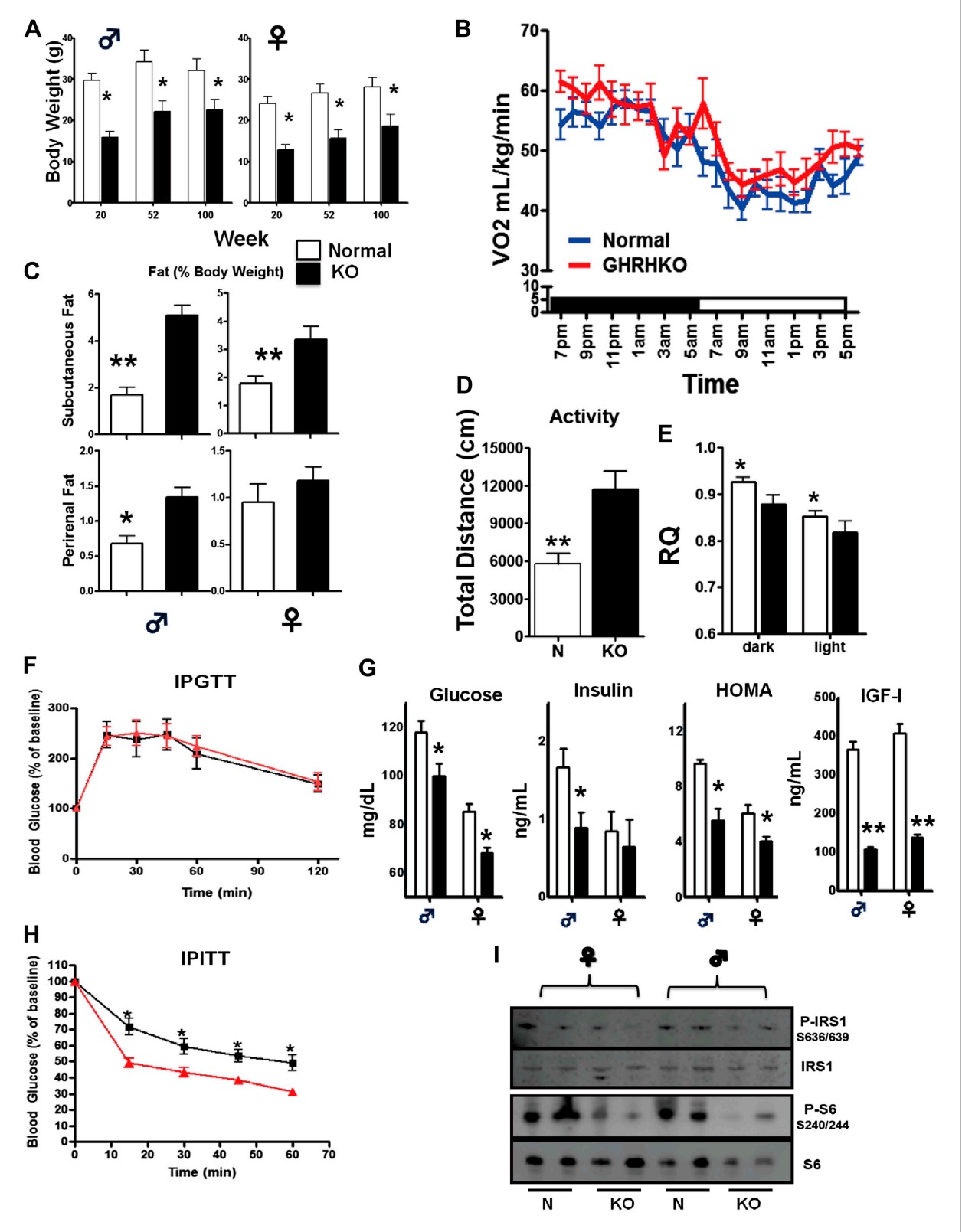

**Figure 2**. Physiological characteristics. (**A**) Body weight. (**B**) Respiratory exchange ratio (VCO$_2$/VO$_2$). (**C**) Fat content presented as an absolute values and percentage of body weight. (**D**) Physical activity. (**E**) RQ values plotted as 12-hr averages representing either dark or light periods on both fed and fasted days. RQ = respiratory quotient; (**F**) Glucose tolerance test (IPGTT). 16 hr-fasted mice underwent GTT by intraperitoneal (i.p.) injection with 1 g glucose
*Figure 2. Continued on next page*

*Figure 2. Continued*

per kg of BW. (**G**) Fasted glucose, plasma insulin, homeostatic model for assessment of insulin resistance (HOMA-IR) and IGF-I levels. (**H**) Insulin tolerance test (IPITT). Mice were i.p. injected with 1 IU porcine insulin per kg of BW. (**I**) Representative immunoblot of the indicated proteins from liver lysates. Each lane corresponds to a different mouse. For graph **B,D,E,F,H**, only data from male mice were shown. N = 8–10/group; each bar represents means ± SEM for 8–10 mice of each group. *p<0.05, **p<0.01, ***p<0.001.

The following figure supplements are available for figure 2:

**Figure supplement 1**. Phosphorylation of S6 in the white adipose tissue (WAT) and muscle (MUS) of GHRH-KO and normal (little-mate control) mice.

(GO:0045595) (p<0.01) (*Figure 3—figure supplement 1*). Increased genes were also disproportionately associated with KEGG pathways, including 'metabolism of xenobiotics by cytochrome P450' and 'drug metabolism-cytochrome P450' (p<0.04). Among the 164 decreased genes, there was overrepresentation of genes associated with oxidation-reduction process (GO:0055114), xenobiotic metabolic process (GO:0006805), monocarboxylic acid metabolic process (GO:0032787), and lipid metabolic process (GO:0006629) (p<0.01) (*Figure 3—figure supplement 1*). KEGG pathways overrepresented among decreased genes included 'steroid hormone biosynthesis', 'metabolism of xenobiotics by cytochrome P450' and 'drug metabolism—cytochrome P450'.

We compared GHRH-KO-increased and GHRH-KO-decreased genes to genes altered by CR in 13 experiments in which hepatic gene expression was compared between CR and ad lib-fed mice (*Figure 3—figure supplement 1*). Surprisingly, GHRH-KO-decreased genes compared more favorably with CR than GHRH-KO-increased genes. For 11 of 13 CR experiments, the 164 GHRH-KO-decreased were biased towards CR-decreased expression, and this trend was significant with respect to 7 of 13 CR experiments (p<0.05 by rank-based GSEA; *Figure 3—figure supplement 1*). However, we only identified 3 of 13 CR experiments in which the 141 GHRH-KO-increased were biased towards CR-increased expression (p<0.05 by GSEA). Hepatic gene expression patterns in GHRH-KO mice were therefore in partial agreement with those of CR, with stronger concordance among GHRH-KO-decreased genes as compared to GHRH-KO-increased genes (*Figure 3—figure supplement 1*).

We next compared GHRH-KO-increased and -decreased genes to results from a broader set of experiments (*Figure 3A,B*). This revealed strong correspondence between genes altered in GHRH-KO mice and those altered in other long-lived mouse models. For instance, *Cyp2b13* and *Igfbp1* were among the genes most strongly elevated in GHRH- KO mice, and expression of both genes was similarly increased in *Ames* (mixed background) (*Amador-Noguez et al., 2004*), *Snell* (DW/J Pit1dw × C3H/HeJ Pit1dw-J background) (*Boylston et al., 2006*), *Little* (Ghrhrlit/lit, B6 background) (*Amador-Noguez et al., 2004, 2007*), *Ghr−/−* (B6 background) (*Rowland et al., 2005*) and *Fgf21* Tg mice (B6 background) (*Zhang et al., 2012*). Conversely, genes most strongly decreased in GHRH-KO mice were also decreased in *Ames, Snell, Little, Ghr−/−* and *Fgf21* Tg mice (e.g., *Hsd3b5, Slco1a1, Igf1, Elovl3, Keg1* and *Igfals*). Consistent with activation of Nrf2 signaling in GHRH-KO mice, genes altered most strongly in GHRH-KO mice were often similarly altered in mice carrying a liver-specific Keap1-KO mutation (*Yates et al., 2009*) (e.g., see *Sult1e1, Spink3, Cyp39a1, Hsd3b5, Slco1a1* and *Igf1*; B6 background; *Figure 3B*). We also observed notable trends in which the effects of GHRH-KO on liver gene expression were correspondent with a feminization of gene expression patterns, as well as with the effects of castration in male mice (A/JCr background; *Figure 3B*) (*Rogers et al., 2007*). Finally, we observed an unexpected trend, in which genes increased and decreased in GHRH-KO mice were similarly altered in mice with streptozocin-induced diabetes (BALB/c background; *Figure 3B*) (*Kobori et al., 2009*).

We speculated that the shifts in hepatic gene expression in GHRH-KO mice could, at least in part, be due to the activation or repression of key transcription factors. To test this possibility, we scanned 2 kb regions upstream of increased and decreased genes for matches to sites within a dictionary of 1291 motifs associated with known DNA-binding proteins. We then used semi-parametric generalized additive logistic models to identify motifs present at significantly elevated frequency among GHRH-KO-altered genes, as compared to all other hepatic related genes. Among the 141 increased genes, the dominant trend was enrichment of hepatocyte nuclear factor (HNF) motifs in upstream regions (*Figure 3C*). The strongest enrichment was observed for an HNF recognition site with consensus sequence 5-GTTAAT-ATT-3 (p=7.3e-10; *Figure 3D*). Among the 164 decreased genes, enrichment for motifs associated with several DNA-binding proteins was observed (*Figure 3C*). However, the strongest

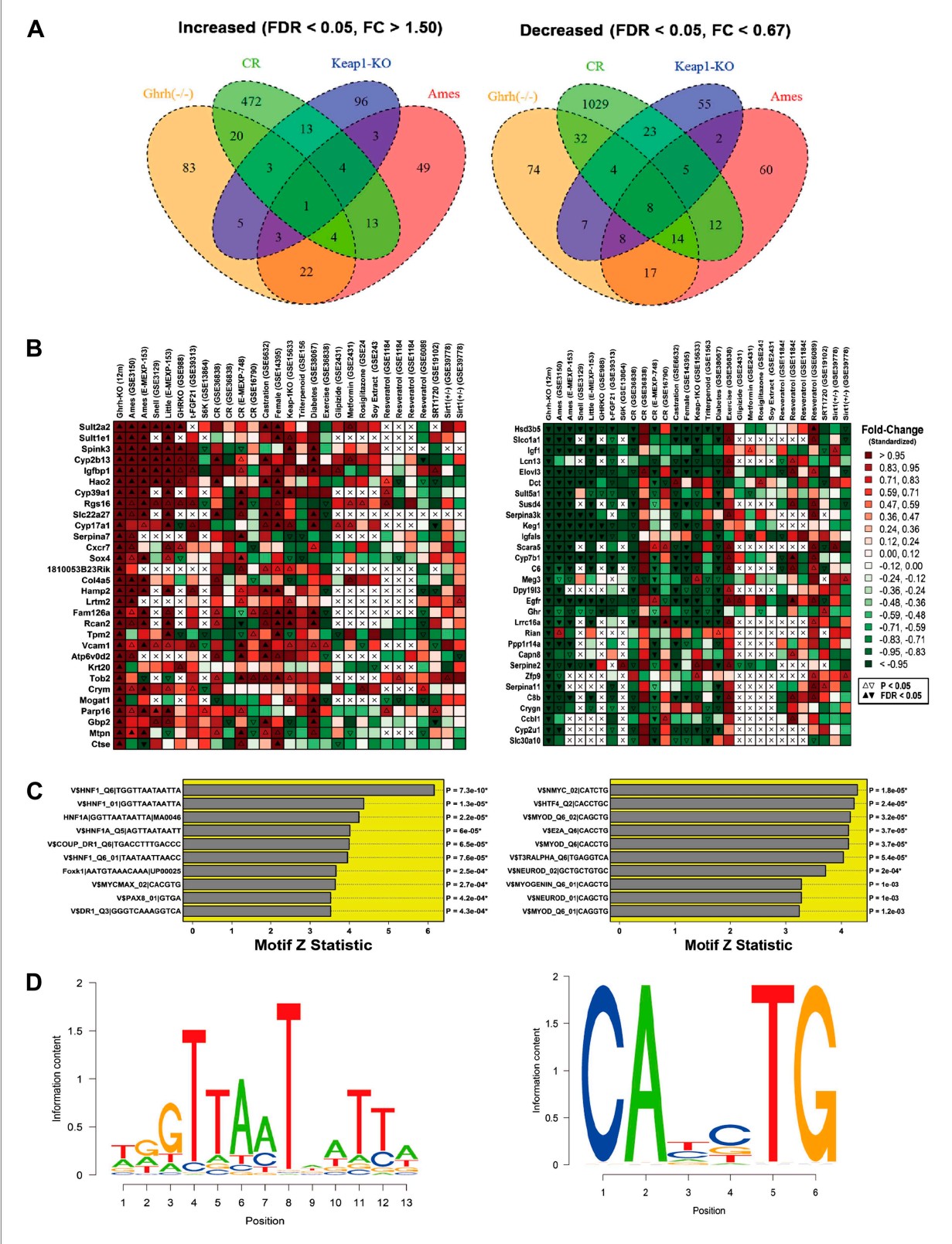

**Figure 3**. Hepatic gene expression profiles in GHRH-KO mice. Microarray analysis was used to identify 141 genes with increased expression in GHRH-KO liver tissue (FDR < 0.05 and FC > 0.05) and 164 genes with decreased expression in GHRH-KO liver (FDR < 0.05 and FC > 0.05). (**A**) Venn diagram showing overlap of increased and decreased genes with sets of genes similarly altered in hepatic tissue of CR-fed mice, mice with a liver-specific Keap-1

*Figure 3. Continued on next page*

*Figure 3. Continued*

mutation, and long-lived Ames dwarf mice. The same Affymetrix platform was used in each experiment (Mouse Genome 430 2.0 array). (**B**) Genes most strongly increased in GHRH mice (left) and genes most strongly decreased in GHRH mice (right). For comparison, heatmap colors show the fold change for each gene in other mutant mouse models and mice provided various treatments (e.g., CR, resveratrol, etc). Liver tissue was evaluated in all cases. (**C**) Top ranked motifs enriched in 2 kb regions upstream of GHRH-KO-increased genes (left) and GHRH-KO-decreased genes (right). Asterisk symbols denote those motifs remaining significant following FDR-adjustment for multiple testing among the 1291 binding sites included within our motif dictionary. (**D**) Sequence logos for the top-ranked motif among GHRH-KO-increased genes (left, V$HNF1_Q6|TGGTTAATAATTA) and the top ranked motif among GHRH-KO-decreased genes (right, V$NMYC_02|CATCTG).

The following figure supplements are available for figure 3:

**Figure supplement 1**. Characterization of genes altered in GHRH-KO mice.

trend was increased frequency of a MYC binding site in upstream regions, with consensus sequence 5-CA—TG-3 (p=1.8e-5; *Figure 3D*). In agreement with this trend, we also noted enrichment of MYC-MAX dimer binding sites upstream of GHRH-KO-increased genes (p=2.7e-4; *Figure 3C*). Genes altered in GHRH-KO liver were thus characterized by increased frequency of HNF and MYC motifs in upstream regions.

## Activation of genes involved in xenobiotic detoxification metabolism in the liver of GHRH-KO mice

Detoxification and elimination of xenobiotics and endobiotics is a major function of the liver and is important for maintaining the metabolic homeostasis of the organism (*Osterreicher and Trauner, 2012*). Recent studies in long-lived C. *elegans* have linked the up-regulation of xenobiotic pathways with increased longevity (*McElwee et al., 2004*). This link has been further supported in mammals, given that several mouse models of delayed aging are characterized by stress resistance and increased xenobiotic gene expression (*Amador-Noguez et al., 2007*; *Steinbaugh et al., 2012*). Consistent with this idea, our microarray analysis showed robust elevation in the expression of genes associated with xenobiotic detoxification pathways (*Figure 3*; *Supplementary file 1A,B*). To confirm the array results, and to gain further insight into the xenobiotic gene regulation patterns in GHRH-KO mice, we used real-time RT-PCR to compare the expression levels of a set of phase I and phase II xenobiotic detoxification genes in liver and small intestine of GHRH-KO vs control mice. Of 15 such phase I mRNAs evaluated, 10 were found to be elevated in liver of KO mice, and seven of these were dramatically elevated compared to controls (*Figure 4A*). In contrast, most of the hepatic phase II genes were modestly increased in GHRH-KO mice (*Figure 4B*). However, *Sult2a2*, the gene increased most strongly in microarray screening, was shown by RT-PCR to be elevated by more than 1000-fold in KO mice (*Figure 4B*). Surprisingly, expression of phase I and II genes was similar in small intestine of KO and control mice, suggesting that the effect of GH on these genes may be liver-specific.

## Treatment with GH reverses the effects of hypopituitarism on hepatic expression of xenobiotic genes

Our data suggested that the disruption of GH signaling leads to elevated expression of genes involved in hepatic xenobiotic metabolism. Previously, we showed that a short-term GH treatment (6 weeks) in *Ames* dwarf mice during early stages of postnatal life could both shorten lifespan and decrease the cellular stress resistance (*Panici et al., 2010*). To determine whether GH replacement also influences the expression of hepatic xenobiotic genes, we evaluated each of these mRNAs in liver of *Ames* dwarf mice that had been exposed to GH replacement for a period of 6 weeks. As shown in *Figure 4C*, consistent with the pattern in GHRH-KO mice, hepatic expression of these genes was robustly elevated in *Ames* dwarf mice compared to their littermate controls (p<0.01). GH-treatment of these mutant mice dramatically suppressed the elevation of these genes including *Cyp2b9, Cyp2b10, Cyp4a14, Fmo3* and *Sult2a2* (two-tailed *t* test; p<0.01) (*Figure 4C*), but had no effect on housekeeping control genes. These findings suggest that GH is a direct regulator of xenobiotic mRNAs in hepatocytes and that elevation of such genes in GH deficient mice is a direct consequence of attenuated GH signaling.

## Different signaling pathways may mediate effects of GHRH deletion

The GH/IGF-1 signaling pathways have been shown to play key role in controlling stress resistance, lifespan, and aging in different organisms (*Kenyon, 2010*; *Bartke, 2011*). We hypothesized that the

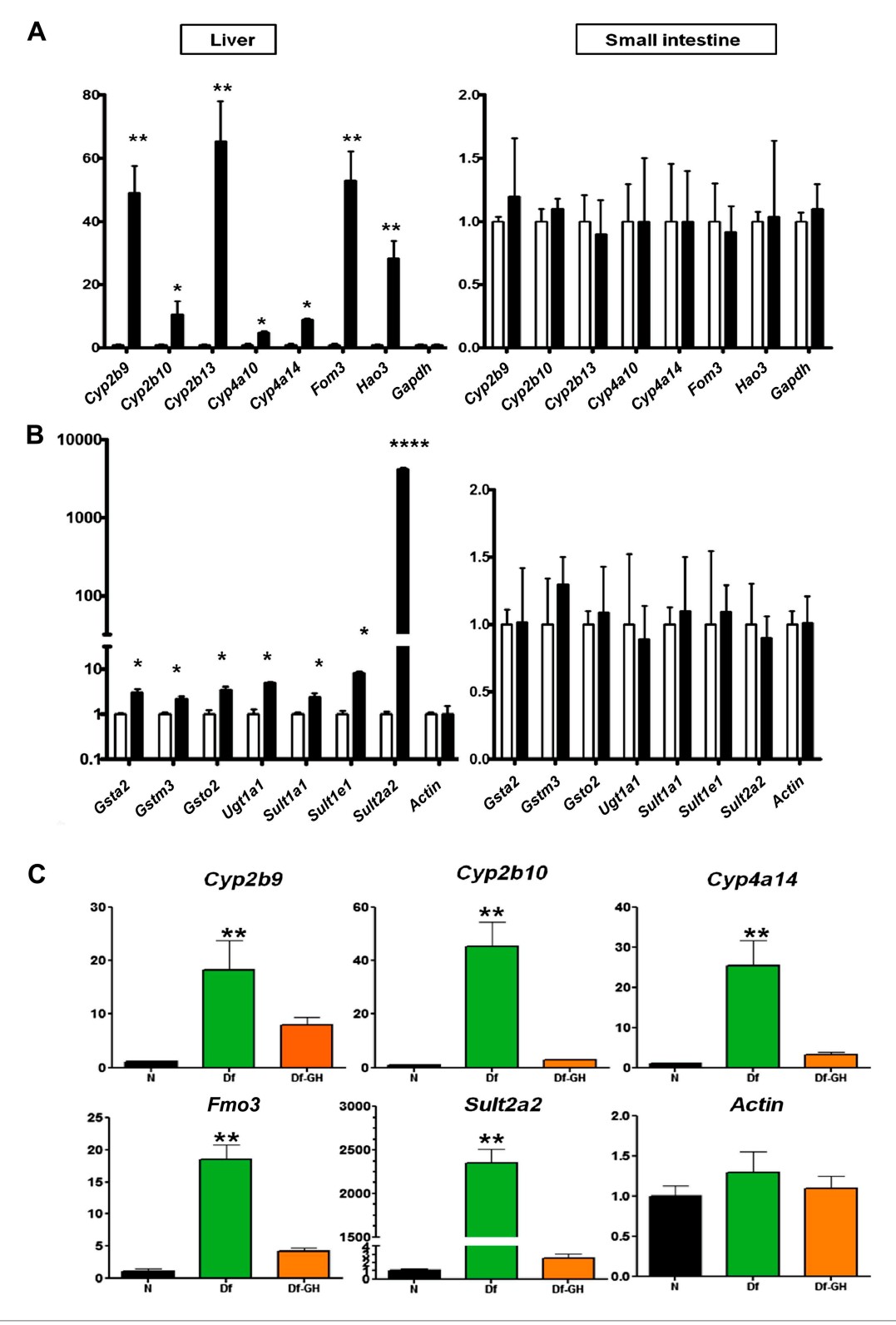

**Figure 4**. Alteration in Xenobiotic detoxification genes. (**A**) Expression of phase I xenobiotic metabolism levels in liver and small intestine of KO and control mice. (**B**) Expression of phase II xenobiotic genes. (**C**) GH replacement effects on hepatic xenobiotic gene expression in GH deficient Ames dwarf mice (df) and littermates. mRNA was
*Figure 4. Continued on next page*

*Figure 4. Continued*

measured using real-time RT-PCR. Data normalized to Gapdh or actin values and were expressed as a ratio (fold change) to levels of mRNA in control mice. Bars indicate mean ± SEM for male KO or df and age-matched control male mice. N = 8 mice per group; *p<0.05, **p<0.01, ***p<0.001.

The following figure supplements are available for figure 4:

**Figure supplement 1**. *Cyp2b10* protein expression is elevated in GHRH-KO mouse liver tissue.

**Figure supplement 2**. Increased cytochrome P450 activity in the liver of GHRH-KO mice.

local action of GH/IGF-1 within specific tissues underlies heightened stress cellular stress responses in GHRH-KO mice.

The family of insulin-like growth factor (IGF) and related molecules comprises growth factors (IGF-I, IGF-II), their receptors (IGF-IR, IGF-IIR), and six structurally related IGF binding proteins (IGFBP-1–6) (*Hwa et al., 1999*). Using real-time RT-PCR analysis, we found a profound suppression of IGF-I transcription and significant up-regulation of *Igf2*, *Igfbp1* and *Igfbp2* mRNA levels in the liver of GHRH-KO mice relative to controls (*Figure 5A*). However, such differences were absent in different brain regions, such as cerebral cortex (*Figure 5A*). This is consistent with previous reports in Ames dwarf mice, indicating that localized regulation of GH/IGF-1-associated mRNAs in the brain is independent of pituitary-derived GH secretion (*Sun et al., 2005*).

We further explored the downstream targets/effects of local IGF-I signaling in liver and brain. In comparison to littermate controls, GHRH-KO mice were characterized by decreased hepatic phosphorylation of MAPKs, including the MEK, ERK, and P38 kinases, each of which is known to participate in cellular stress responses (*Figure 5B*). In contrast, brain tissues from GHRH-KO mice showed no change in phosphorylation of these kinases (*Figure 5D*). These data further support preservation of local GH/IGF-I signaling in brain regions of KO mice.

The transcriptional regulation of immediate early genes (IEGs) including *Egr1*, *Fos*, and *Jun* are dependent on activation of MAPK signaling pathways. As expected, hepatic IEGs mRNA levels are significantly repressed in GHRH-KO mice (*Figure 5C*) whereas no alteration was detected in the brains.

Our microarray analysis uncovered correspondence between the hepatic gene expression profiles of GHRH-KO mice and mice carrying a liver-specific Keap1-KO mutation, suggesting that Nrf2 activity might be elevated in GHRH-KO mice (*Figure 3*). We thus examined Nrf2 signaling in GHRH-KO tissues. As shown in *Figure 5E*, Western blot analysis of liver lysates showed that GHRH-KO mice had a higher nuclear level of Nrf2 protein than control animals. To further assess Nrf2 activity in the liver, we next evaluated expression of Nrf2 dependent genes by quantitative RT-PCR (*Figure 5F*). These genes included those encoding glutamate cysteine ligase modifier subunit 1 (GCLM), quinoneoxidoreductase 1 (NQO1), thioredoxinreductase (TXNRD), NAD(P)H quinoneoxidoreductase 1 (NQO1) and metallothioneins 1 and 2. Each of these Nrf2-dependent genes was expressed at higher levels in liver from KO mice than in liver from controls. This further supports up-regulation of Nrf2 signaling in GHRH-KO mice.

## Response of GHRH-KO mice to CR

Our microarray analysis of gene expression indicated that disruption of the GHRH gene had effects that were only partly overlapping with those of CR, indicating that isolated GH deficiency and CR may extend lifespan by independent pathways.

To evaluate the effects of CR, 199 GHRH-KO mice and 211 of their normal siblings were divided into two groups after ad libitum (AL) feeding for the first 12 weeks of life, and then were subjected either to CR or to continued AL feeding. CR produced the expected reduction in body weight in both GHRH-KO and control mice (*Figure 6B,C*). The relative decrease in body weight in response to CR was comparable between KO and control mice in both sexes. The convergence of average body weighs of N-AL and N-CR males after 80 weeks of age almost certainly represents age-related weight loss in N-AL animals and its delay in the N-CR group.

Kaplan–Meier survival curves (*Figure 6A*) indicate that in GHRH-KO mice, CR produced significant increase in overall survival, average, median, and maximal longevity when the data from both sexes were combined (log-rank test, p<0.0001). Further analysis of each sex separately showed that CR

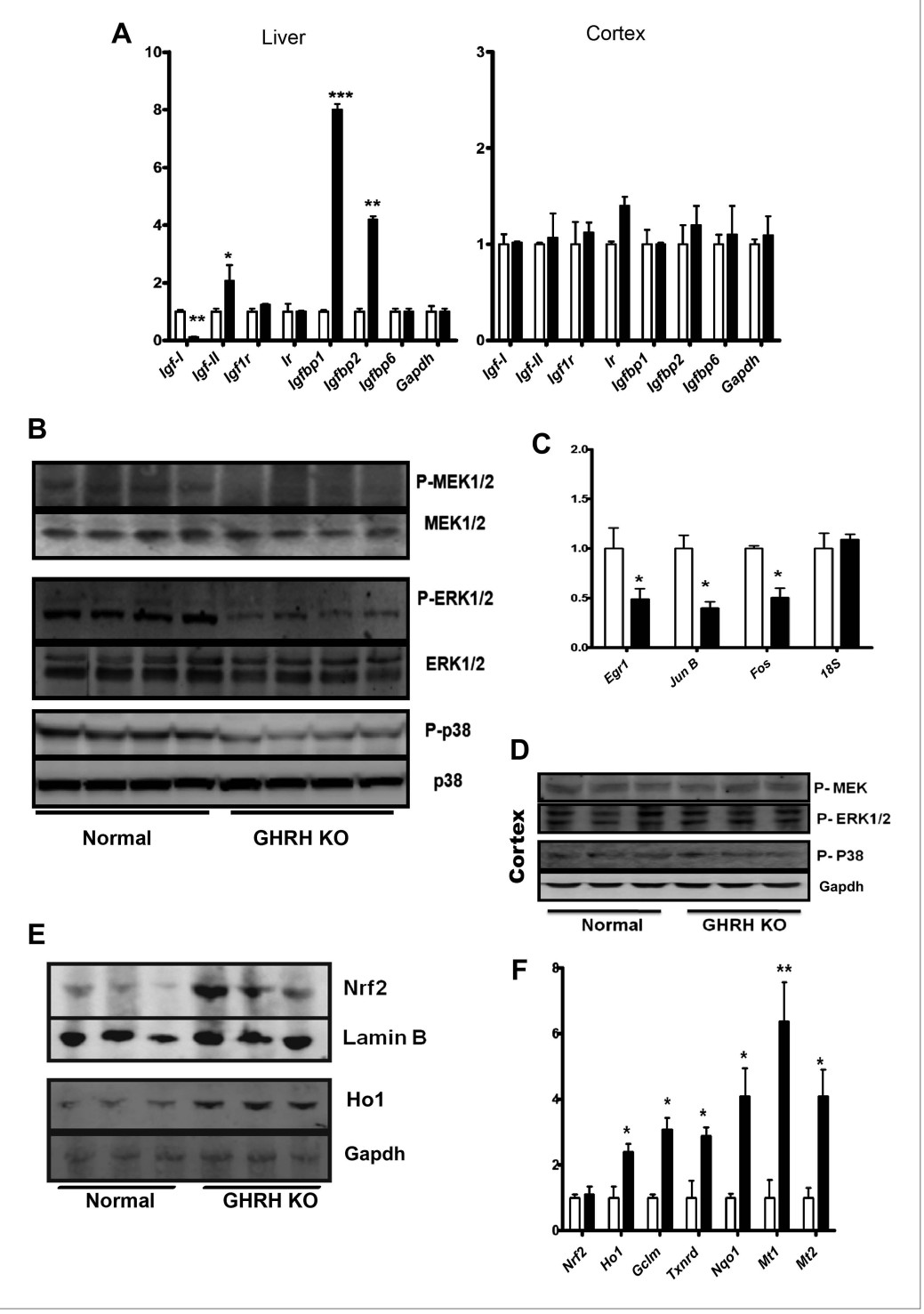

**Figure 5**. Stress signaling pathways. (**A**) Expression of IGF family related mRNA levels in the liver and cortex of GHRH-KO and control mice using real-time RT-PCR. Data are normalized to Gapdh values and expressed as a ratio to the level seen in control mice. Each bar represents means ± SEM for eight mice of each genotype, *p<0.05, **p<0.01, ***p<0.001. (**B**) Representative autoradiographs of Western blots for phosphorylated and total forms of MEK, ERK1/2 and P38 protein in liver lysates of KO and control mice. Each lane corresponds to a different mouse. (**C**) Expression of *Egr1*, *Jun* and *Fos* mRNA levels in the liver KO and control mice. Data are normalized to 18S values and expressed as a ratio to the level seen in control mice. (**D**) Representative immunoblot of the indicated proteins from cortex lysates. (**E**) Representative autoradiographs of Western blots detecting nuclear accumulation

*Figure 5. Continued on next page*

*Figure 5. Continued*

of Nrf2 and Ho1 in the cytoplasm from liver lysates. Gapdh was the cytoplasmic marker and Lamin B was the nuclear marker. Each lane corresponds to a different mouse. (**F**) Expression of Nrf2-dependent mRNA levels in the liver KO and control mice by mRNA Q-PCR. Data are normalized to Gapdh values and expressed as a ratio (fold change) to the level seen in control mice. N = 8 male mice per group; *p<0.05, **p<0.01, ***p<0.001.

female KO mice out-live AL KO mice by 21% (median lifespan: 1156 days vs 956 days; p=0.0001; logrank test; p<0.0001; *Figure 6E*). In contrast, the median lifespan of male CR KO mice does not significantly exceed that of male AL KO mice (945 days vs 928 days *Figure 6D*), but CR still further increased overall survival of male KO mice (logrank test; p=0.0382). As expected, the CR-induced increase in longevity in the control mice was similar to findings in other strains of mice (p<0.0001). Applying the method of Wang/Allison, maximal lifespan of KO mice was increased by CR in both sexes, by 17% for females (p<0.05; *Figure 6E*) and by 9% for males (p<0.05; *Figure 6D*).

These results, taken together, support the idea that mechanisms underlying extended longevity in GHRH-KO mutants and CR-fed mice are at least partially distinct.

## Physiological parameters: differential responses to CR between GHRH-KO and control mice

Fasting plasma glucose levels were significantly reduced in GHRH-KO AL-fed mice compared with control AL-fed mice (p<0.005; *Figure 7A*); CR did not significantly affect the glucose level in either genotype. Plasma insulin was greatly decreased in N-CR, KO-AL, and KO-CR relative to N-AL. CR also resulted in significant further reduction of insulin levels in GHRH-KO mice compared with their AL group (p<0.05). As expected, CR resulted in a significant reduction in plasma IGF-I levels in controls, whereas IGF-I levels in GHRH-KO mice remained at similar very low levels regardless of their diet (*Figure 7A*). The level of adiponectin, a plasma protein secreted specifically from adipocytes, was higher in GHRH-KO than in the control group (p<0.01; *Figure 7A*). Interestingly, CR significantly increased plasma adiponectin levels in both genotypes (p<0.005 and 0.01 respectively; *Figure 7A*). Conversely, circulating leptin levels were reduced by CR only in KO mice (p<0.01), while KO AL animals had higher leptin levels than the controls (p<0.05; *Figure 7A*). Moreover, fasting plasma triglyceride and cholesterol levels were significantly decreased in KO mice and further suppressed by CR (p<0.05; *Figure 7A*).

Fibroblast growth factor 21 (FGF21), a critical metabolic regulator of glucose and lipid metabolism, is secreted primarily from the liver in response to prolonged fasting and plays a major role in coordinating adaptive changes such as mobilizing and burning fatty acids among different tissues (*Potthoff et al., 2012*). A most recent study has shown that overexpression of FGF21 markedly extends lifespan in mice (*Zhang et al., 2012*). Intriguingly, the Fgf21 transgenic mice have been found to be acting to suppress GH action (*Inagaki et al., 2008*). In this context, we further evaluated the effect of GHRH deletion and CR on FGF21 expression. To our surprise, CR profoundly suppressed serum FGF21 levels in both male and female GHRH-KO and control mice (p<0.001; *Figure 7B*) whereas isolated GH deficiency had almost no effect on FGF21 levels. Since liver is the major source of circulating FGF21, hepatic FGF21 mRNA expression was quantified by real-time RT-PCR. Consistent with plasma levels, the transcriptional levels of FGF21 in the liver were markedly inhibited by CR in both GHRH-KO and control mice (*Figure 7C*). However, no change was detected in the levels of the receptor FGF4R and β-Klotho in liver (*Figure 7—figure supplement 1*).

## Discussion

The number of mutations known to increase mouse lifespan has increased considerably over the last decade, although often the degree of lifespan increase has been modest in magnitude (*Swindell, 2009*). The largest increases have been observed in the *Ames* and *Snell* dwarf mice (e.g., 40–50% increase) and these lifespan effects have been replicated in several laboratories with different mouse strains (*Bartke, 2011*). However, deficiencies of multiple hormones in these models might confound the direct and causal effects of GH deficiency.

In an attempt to overcome this difficulty, we previously found that early life GH (but not thyroxine) replacement almost completely restored Ames dwarf lifespan to the level of wild-type littermate controls (*Panici et al., 2010*). The little mice carrying a naturally occurring missense mutation in the

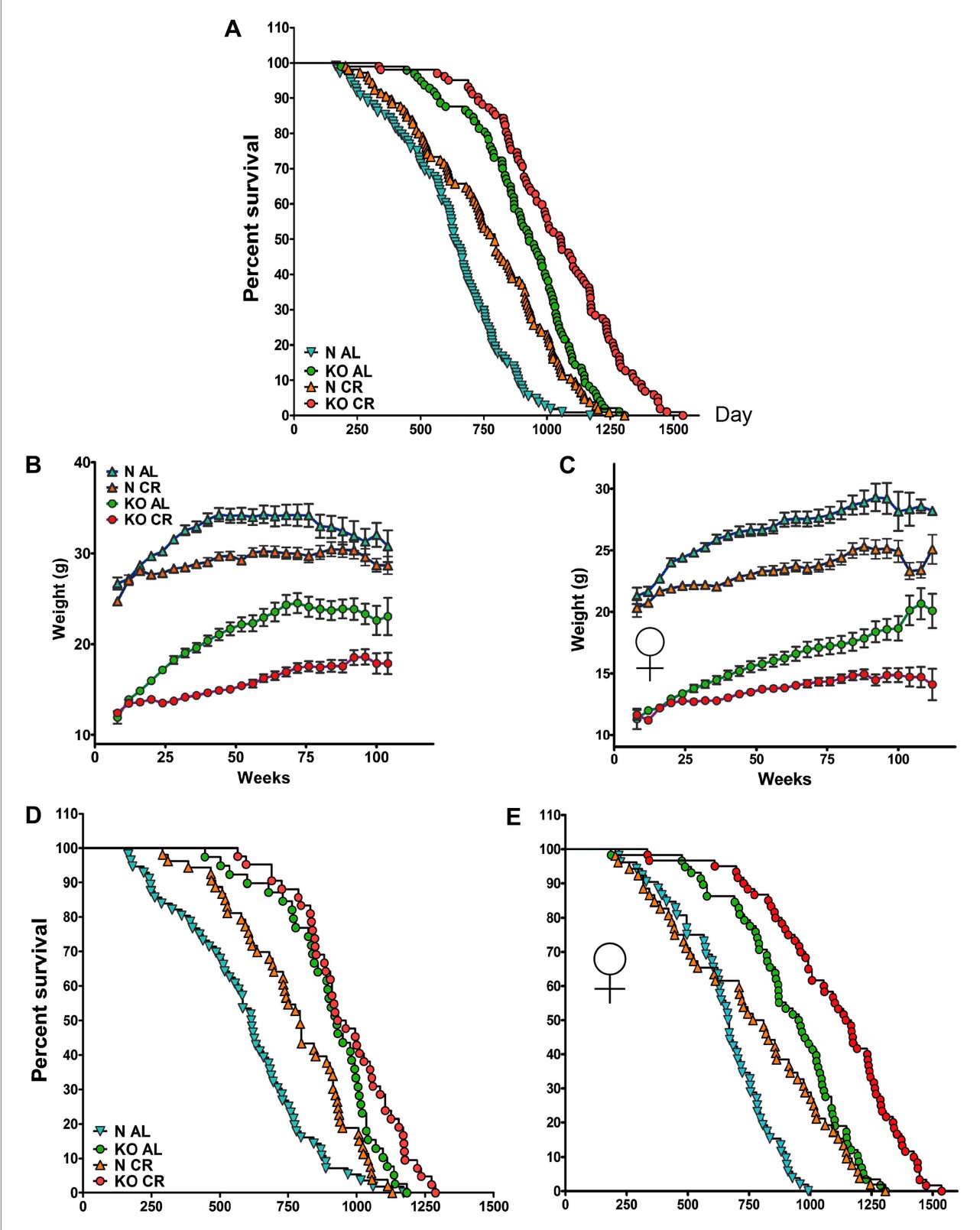

**Figure 6**. Lifespan and growth curve in response to long-term CR. (**A**) Kaplan–Meier survival plot of GHRH-KO (KO) and Control mice (N) that were fed AL or subjected to long-term CR (Sex pooled); N-AL (N = 108), N-CR (N = 105), KO-AL (N = 97) and KO-CR (N = 102). Time course of changes in body weight for male (**B**) and female (**C**) GHRH-KO and control mice that were fed AL or subjected to CR. Animals were weighed weekly starting at 4 weeks of
*Figure 6. Continued on next page*

*Figure 6. Continued*

age. (**D**) Kaplan–Meier survival plot of male KO and Control mice that were fed AL or subjected to long-term CR; N-AL (N = 56), N-CR (N = 53), KO-AL (N = 39) and KO-CR (N = 42). (**E**) Survival plot of female KO and Control mice; N-AL (N = 52), N-CR (N = 52), KO-AL (N = 58) and KO-CR (N = 60).

GHRH receptor (*Ghrhr^lit/lit*) have isolated GH deficiency (*Eicher and Beamer, 1976*; *Godfrey et al., 1993*). While increased lifespan was also reported in the *little* model, the longevity phenotype is less substantial and depends on a relatively low fat content of the diet employed (*Flurkey et al., 2001*). In the current study, we find that a robust increase in lifespan in both male and female GHRH-KO mice. Together, our results provide a compelling evidence for the idea that GH pathway is the major regulatory pathway for longevity in mice.

The remarkable magnitude of the extension of longevity in GHRH-KO mice may be related in some ways to the relatively short lifespan of normal animals from this strain, approximately 21 months, as compared to approximately 24 months in normal siblings of *Ames* dwarfs and approximately 29 months in normal siblings on *Ghr−/−* mice in our colonies (*Brown-Borg et al., 1996*; *Bartke et al., 2001*; *Bonkowski et al., 2006*). Studies in *Ghr−/−* mice indicate that the presence of a significant extension of lifespan in animals lacking GH signals does not depend on genetic background but the magnitude of this effect tends to be reduced in a long-lived strain (*Coschigano et al., 2003*). Differences in the genetic background of GHRH-KO mice and *Ames* dwarf, *Snell* dwarf, and *Ghr−/−* mice used in previous studies in different laboratories make it difficult to determine whether absence of GHRH, GH or their receptors have quantitatively similar or differential impacts on longevity. These comparisons are further complicated by differences between the diets used in different laboratories. Nevertheless, the increased longevity we observed in GHRH-KO animals is a robust finding when compared with same genetic background controls.

## Potential mechanisms mediating reduced GH signaling with extended longevity

Improved insulin sensitivity combined with low insulin and glucose levels are characteristics of a series of long-lived mutant mice and mice with long-term CR (*Bartke, 2011*). On the contrary, opposite phenotypic features characteristic of metabolic syndrome, including hyperinsulinemia, and insulin resistance, are associated with increased risk of various age-related diseases and with reduced life expectancy (*Stensvold et al., 2011*; *Noale et al., 2012*). Consistent with this notion, insulin sensitivity is also improved in the GHRH-KO mice, further supporting that this is a key contributor of lifespan extension of mice. We suspect that at least some of the gene expression responses observed in hepatic tissue of GHRH-KO mice can be attributed to enhanced insulin signaling. For instance, the GHRH-KO-increased genes we identified included Sox4 and Vcam1, each of which have been associated with response to glucose stimulus. In addition, among the 164 GHRH-KO-decreased, upstream regions were enriched with motifs recognized by MYC, which is a transcription factor associated with the TOR branch of the insulin signaling pathway.

Increased resistance to multiple forms of lethal stress has been consistently associated with lifespan extension in long-lived mutant worms, flies and mice suggested the appealing idea that the longevity seen in these mutants was a consequence of their stress resistance (*Miller, 2009*). Our evaluation of changes in hepatic gene expression profiles and stress response pathways also points toward this direction. For instance, several of the genes with increased expression in GHRH-KO liver were associated with the GO biological process term 'cellular response to stress' (e.g., Blm, Ccnd1, Dclre1a, Ddx1, Gpx3, Il1rn, Krt20, Morf4l2, Ppargc1a, Prpf19, Rtn4, Slc2a1, Sox4 and Vldlr). Additionally, we noted that several genes with increased expression in GHRH-KO liver had also been induced in experiments where mice had been exposed to injury or toxicity stressors, including partial hepatectomy (e.g., Cenpq, Ccnd1 and Cdkn2c), the HSF-activating compound quercetin (e.g., Gpr146, Il1rn and Tef), and high doses of acetaminophen (e.g., Vcam1, Mafb and Slc2a1).

The transcription factor Nrf2 plays a central role in regulating many antioxidant-related genes against multiple types of stress (*Ma, 2013*). Under normal conditions, Nrf2 is sequestered in the cytosol by Keap1, the cytosolic repressor of Nrf2. In response to various stress, Nrf2/Keap1 binding is disrupted, and Nrf2 translocates into the nucleus to stimulate the activation of various antioxidant-associated genes (*Ma, 2013*). Overexpression of SKN-1 (Nrf2 ortholog) can prolong

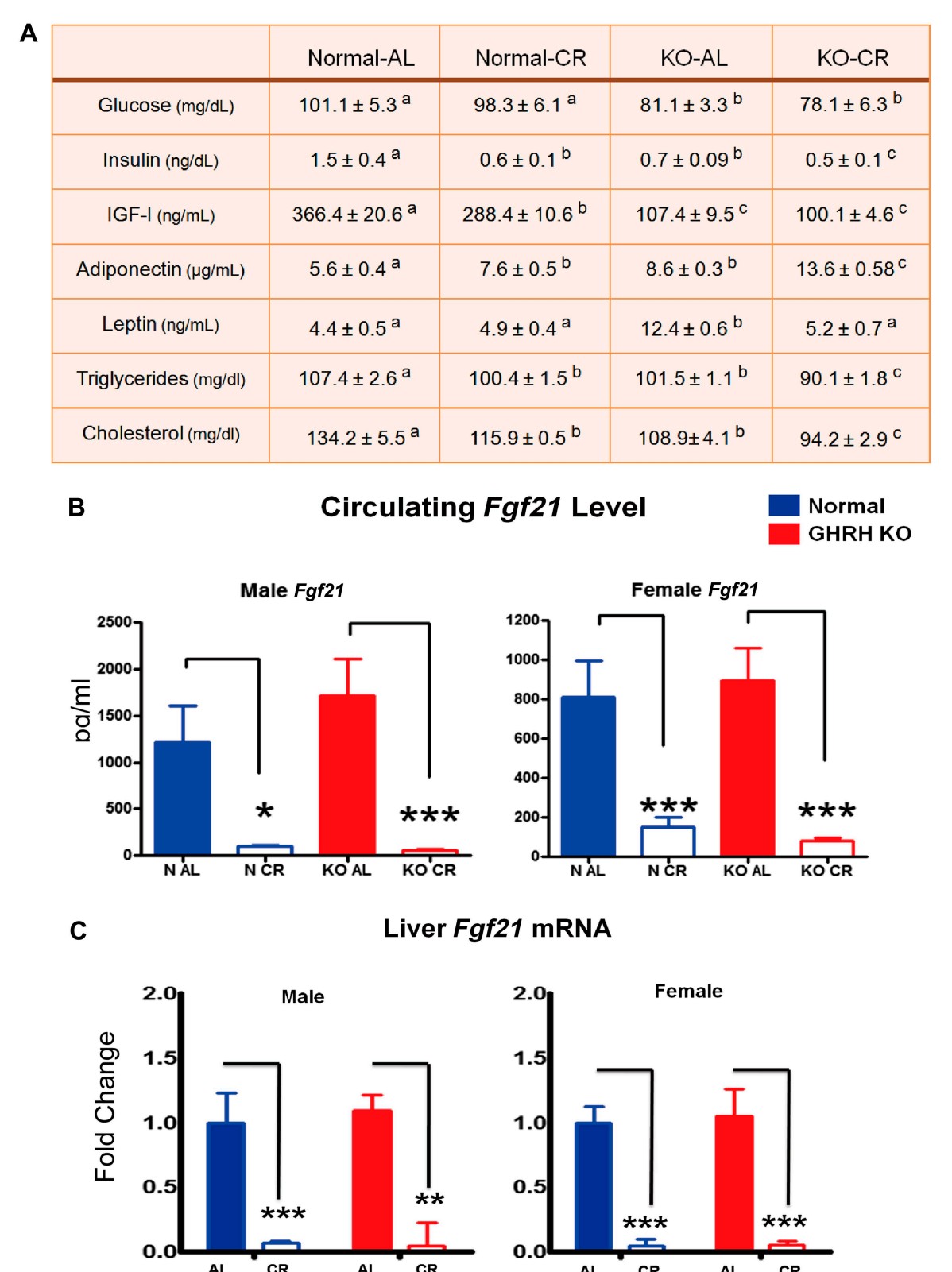

**Figure 7**. Blood parameters alteration in response to CR. (**A**) Various plasma parameters from GHRH-KO and Control male mice subjected to caloric restriction. Different superscripts denote significant difference at p<0.05. Data represent the means ± SEM. (**B**) Circulating levels of Fgf21. (**C**) Hepatic

*Figure 7. Continued on next page*

*Figure 7. Continued*

Fgf21 mRNA levels. Data normalized to Gapdh or actin values and were expressed as a ratio to levels of mRNA in control mice. Bars indicate mean ± SEM. N = 8 mice per group; *p<0.05, **p<0.01, ***p<0.001.

The following figure supplements are available for figure 7:

**Figure supplement 1**. Hepatic FGF4R and β-Klotho mRNA levels.

*C. elegans* lifespan (*Tullet et al., 2008*). Disruption of Keap1, which in turn increase Nrf2 activity, also increases the male *Drosophila melanogaster* longevity (*Sykiotis and Bohmann, 2008*). However, whether Nrf2 can contribute to the mammalian aging and affect lifespan remains undemonstrated. Intriguingly, as revealed by our microarray comparison, liver-specific Keap1 knockout mice have increased resistance to hepatic stressor acetaminophen and exhibit a very similar hepatic gene expression pattern to the one of GHRH-KO mice (*Figure 3A*) (*Okawa et al., 2006*). Genes induced in Keap1(−/−) and GHRH-KO mice include Nqo1, Adora1, Cyp39a1, Trim24 and Gstt3, while genes decreased in both models include Ttc39c, Tmem19, Egfr, Mcm10 and Slco1a1. Moreover, nuclear Nrf2 protein and many ARE genes were upregulated in GHRH-KO mice suggesting that Nrf2 could play an important role in lifespan extension of these long-lived mice.

All organisms are continuously exposed to a wide variety of exogenous and endogenous toxic compounds in the environment. Recently, it has been proposed that a generalized up-regulation of xenobiotic detoxification pathways may be a shared feature of long-lived mutants and a key mechanism of longevity assurance (*McElwee et al., 2004*; *Shore et al., 2012*; *Steinbaugh et al., 2012*). Our findings show both a strong increase and decrease in xenobiotic metabolism genes in GHRH-KO mice. Among GHRH-KO-increased genes, several were associated with P450 drug metabolism (e.g., Cyp2b13, Cyp2c38 and Cyp2c39). Similarly, among GHRH-KO-decreased genes, we noted that several were associated with xenobiotic metabolic processes (e.g., Acsl1, Gstp1 and Ugt2b1).

The mechanisms underlying activation of xenobiotic metabolism genes in the absence of environmental toxins remain unknown. Some studies have shown elevation of bile acids (as endogenous xenobiotics) level and altered bile acid metabolism in GH deficient mice (*Amador-Noguez et al., 2004*, *2007*). Moreover, several GH-mediated transcriptional factors including Nrf2, STAT 5b, and hepatic nuclear factor (HNF) 4alpha have been shown to play an important role in xenobiotic gene expressions (*Waxman and Holloway, 2009*; *Wu et al., 2012*). In this study, we postulate that the activation of xenobiotic metabolism pathway, presumably through inhibition of GH signaling and alterations of the corresponding transcriptional factors, protects the tissues from the damaging effects of endogenous and exogenous molecules and thus contributes to maintaining tissue and metabolic homeostasis during aging.

We note that an important avenue for future work will be to carry out end-of-life pathology studies of GHRH-KO mice. This will provide information on likely causes of death and how such causes may differ from those in control animals. At this point, however, based upon findings in other GH-related long-lived mutants (*Ikeno et al., 2003*; *Vergara et al., 2004*; *Ikeno et al., 2009*), we suspect that reduced incidence and/or delayed onset of cancer may have contributed to extended longevity of GHRH-KO mice.

## Different CR responses in long-lived GH deficient and GH resistant mice

Comparison of the impact of CR on longevity in GHRH-KO mice to the results obtained previously in other long-lived GH-related mutants reveals both similarities and differences. Resembling findings in Ames dwarf mice (*Bartke et al., 2001*), CR produced a further significant extension of longevity in both sexes of GHRH-KO animals. In contrast to these findings, in GH resistant Ghr−/− mice, CR does not alter longevity of males and produces only a small, although statistically significant reduction of late mortality in females (*Bonkowski et al., 2006*). Relating these differential responses to CR to the impact of these mutations on insulin levels lends support to our hypothesis that suppression of insulin levels is an important mechanism of life extension by CR. Indeed CR has no or little impact on longevity of animals in which insulin levels are already profoundly suppressed (*Bonkowski et al., 2006*). However, we cannot exclude that differences in the genetic background of animals could also partially explain these differences.

The CR-induced extension of longevity in GHRH-KO mice was greater in females than in males, while in GHRH-KO mice subjected to CR, longevity was increased in females only. The mechanism for the sex dimorphism in response to CR is poorly understood. Intriguingly, deletion of several genes

related to insulin/IGF-I signaling including S6k1 (*Selman et al., 2009*), IRS1 (*Selman et al., 2008*) extends longevity only in females. Perhaps divergent effects of male and female sex hormones on adiposity, adipose tissue distribution, inflammation, cardiac function and/or neuroprotection renders males less responsive to the beneficial effects of reducing insulin, somatotropic and mTOR signaling by dietary or genetic interventions (*Bartke et al., 2013*).

It deserves emphasis that the ablation of GHRH in female mice markedly extends longevity even though insulin levels are not significantly reduced. Insulin levels are reduced in Ames dwarf, Snell dwarf, and Ghr−/−, but not in transgenic mice expressing a GH antagonist in which longevity is not increased (*Coschigano et al., 2003*). Suppression of insulin levels is one of the most consistent and robust responses to CR in mice and in other mammalian species. Accordingly, even in humans, insulin sensitivity seems to be a common feature found in centenarians (*Paolisso et al., 1996*).

Intriguingly, GHRH-KO mice have a significantly higher percent body fat (*Figure 2*) throughout their lifespan. In agreement with the increased obesity, KO mice have elevated serum level of leptin. Interestingly, we found that long-term CR reduced circulating leptin level but increased the adiponectin level in GHRH-KO mice. These results suggest that GH signaling interacts with CR affecting gene expressions and secretion patterns of adipose tissues in these animals.

## GH action, Fgf21 and CR effect

FGF21 plays a major role in eliciting and coordinating the adaptive starvation response and promotes similar physiological changes caused by CR, including decreased glucose levels, increased insulin sensitivity, and improved lipid mobilization (*Potthoff et al., 2012*). A most recent study has shown that overexpressing FGF21 in mice prolongs lifespan dramatically (*Zhang et al., 2012*). Furthermore, Fgf21 Tg and CR-fed mice show similarity with respect to their hepatic gene expression profiles (*Zhang et al., 2012*). For these reasons, FGF21 is speculated to increase longevity by partially mimicking the effect of CR in liver and potentially other tissues. In previous studies, however, CR did not induce transcription of hepatic FGF21, nor did CR alter levels of FGF21 in circulation (*Sharma et al., 2012*; *Zhang et al., 2012*). Intriguingly, our data show that long-term (over 12 months) 40% CR profoundly suppressed hepatic FGF21 mRNA and plasma FGF21 concentrations in both GHRH-KO and control mice. This result differs from previous studies, potentially due to differences in mouse strains or the use of short-term CR in earlier work. Nevertheless, these results do not support FGF21 as a key mediator of the pro-longevity effects of CR.

Previously, FGF21 has been shown to cause GH resistance and inhibit hepatic GH signaling through Stat5-dependent manner (*Inagaki et al., 2008*). Interestingly, we found no alteration in FGF21 expression in GHRH-KO mice compared to controls. Moreover, our hepatic transcriptome analysis has revealed pretty good overlap in genes affected by GHRH deletion and FGF21 overexpression (*Figure 3B*). Thus, our results further support the idea that FGF21 longevity effect is likely mediated by the GH signaling. Also, the regulatory crosstalk between GH signaling and FGF21 action may play a pivotal role in controlling metabolic homeostasis during starvation and chronic dietary restriction.

It will be imperative, in future work, to better understand how CR and GH signaling interact with each other at both the metabolic and cellular levels. Knowledge gained through these experiments will shed light on the mechanism by which aging and lifespan are regulated by the GH pathway.

## Materials and methods

### Animals

GHRH-KO mice and their littermate controls (on a mixed C57BL6 and 129SV background) were bred in a closed colony at the RS's laboratory (*Alba and Salvatori, 2004*), housed under standard conditions (12-hr light/12-hr dark cycling and 20–23°C), and fed Lab Diet Formula 5001 (23% protein, 4.5% fat, 6% fiber) (Nestle Purina, St. Louis, MO). All animals were fed AL for the first ~12 weeks of life. Thereafter, the mice were either fed AL (AL groups) or 40% of AL (CR groups). Mice were weighed approximately 16–20 hr after the CR groups had been fed. Animal protocols were approved by the Animal Care and Use Committee of Southern Illinois University.

### Glucose tolerance test and insulin tolerance test

16 hr-fasted mice underwent GTT by i.p. injection with 1 g of glucose per kg of body weight (BW). Blood glucose levels were measured at 0, 15, 30, 45, 60, and 120 min using a PRESTO glucometer

(AgaMatrix, Salem, NH) for GTT. Non-fasted mice were injected by i.p. with 1 IU porcine insulin (Sigma, St. Louis, MO) per kg of BW Blood glucose levels were measured at 0, 15, 30, and 60 min for ITT. The data for both ITT and GTT are presented as a percentage of baseline glucose. p values were calculated by unpaired, two-tailed Student's $t$ tests to compare the specific time points. The homeostasis model assessment of insulin resistance (HOMA-IR) was calculated using glucose and insulin concentrations obtained after 7 hr of food withdrawal, using the following formula: fasting blood glucose (mg/dl) × fasting insulin (µU/ml)/405.

## Microarray analysis

Hepatic gene expression in GHRH-KO mice and normal littermates was profiled using the Affymetrix Mouse Genome 430 2.0 oligonucleotide array platform (n = 3 per genotype, males, 10 months of age). Standard Affymetrix quality control metrics were calculated for each hybridization, including the percentage of probe sets with signals detected above background (i.e., percent present), global RNA degradation score, average background, intensity scale factor, and four measures derived from the fitting of probe-level models (RLE median, RLE IQR, NUSE median and NUSE IQR). Hierarchical cluster analysis was also performed to identify potential outliers. All chips were retained for further analyses based upon these quality control evaluations. Expression scores were calculated for each chip using robust multichip average (RMA). The Affymetrix Mouse Genome 430 2.0features a total of 45101 probe sets; however, we removed 19956 from consideration since they were not significantly expressed above background in any of the six array hybridizations (Wilxon signed rank test as implemented in the MAS 5.0 algorithm). Our analysis was thus based upon 25145 probe sets with expression significantly above background in at least one of the six hybridizations (13201 unique mouse genes). Probe sets showing increased or decreased expression were identified using empirical Bayes methods as implemented in the limma software package (R Bioconductor). An FDR correction for multiple hypotheses testing among all probe sets was performed using the Benjamini-Hochberg procedure. Since the Mouse Genome 430 2.0 includes some 'sibling' probe sets that target mRNAs associated with the same gene, we limited redundancy in our results by considering only the most significantly altered probe set associated with a given gene (i.e., the probe set with the lowest p value). Overall, we identified 365 genes differentially expressed between GHRH-KO mice and normal littermates (141 increased and 164 decreased genes; FDR < 0.05 with fold-change >1.5 or <0.67). With respect to these gene sets, we tested for significantly overrepresented Gene Ontology and KEGG terms using a conditional hyper-geometric test as implemented in the GOstats package (R Bioconductor). For these analyses, significant overrepresentation was assessed relative to a background gene set that included only the 12836 liver-expressed genes not identified as differentially expressed. Enrichment of motifs in regions upstream of differentially expressed genes was evaluated by screening a dictionary of 1291 motifs associated with DNA-binding proteins, which had been assembled by pooling motifs available from the Jaspar, UniPROBE and Transfac databases. Motif enrichment in regions upstream of differentially expressed genes was assessed using semi parametric generalized additive logistic regression models (*Swindell, 2012*; *Swindell et al., 2012*). Protein-coding sequences and repetitive DNA elements were masked in all sequence scans. In all cases, motif enrichment in regions upstream of differentially expressed genes was assessed relative to a background gene set that included only liver-expressed genes not identified as differentially expressed.

## Real-time RT-PCR

Quantitative real-time PCR was performed using a Rotor-Gene 3000 system (Corbett Research) with a QuantiTect SYBR Green RT-PCR kit (Bio-rad) as described (*Sun et al., 2011*). In brief, cells were homogenized with RNA extraction buffer (TRIZOL reagent; Life Technologies, CA) to yield total RNA following the manufacturer's instructions. Total RNA was reverse transcribed with poly-dT oligodeoxynucleotide and SuperScript II. After an initial denaturation step (95°C for 90 s), amplification was performed over 40–45 cycles of denaturation (95°C for 10 s), annealing (60°C for 5 s), and elongation (72°C for 13 s). Amplification was monitored by measuring the fluorometric intensity of SYBR Green I at the end of each elongation phase. The oligonucleotide-specific primers are shown in *Supplementary file 1C*. Glyceraldehyde-3-phosphate dehydrogenase (Gapdh) or beta-actin expression was quantified to normalize the amount of cDNA in each sample. The change in threshold cycle number (Ct) was normalized to the Gapdh reference gene by subtracting $Ct_{Gapdh}$ from $Ct_{gene}$. The effect of treatment (Ct) was calculated by subtracting $Ct_{normal}$ from $Ct_{Tg}$. Fold induction was determined by calculating $2^{Ct}$.

## Western blot analysis

Tissues were homogenized in 0.5 ml ice-cold T-PER tissue protein extraction buffer (Thermo Scientific, Rockford, IL) with protease and phosphatase inhibitors (Sigma). 40 µg of total protein was separated electrophoretically according to size by SDS-polyacrylamide gel electrophoresis using Criterion XT Precast Gel (Bio-Rad, Hercules, CA), and blotted with the antibodies. For visualization of specific bands in the chemiluminescence assays, the membrane was exposed to X-OMAT film; for chemifluorescence the membrane was incubated with ECF (enhanced chemifluorescence) substrate and a digital image was generated with the Molecular Dynamics Storm system. Quantification of immunoblot signals was performed using the ImageQuant software package (Molecular Dynamics, Sunnyvale, CA).

## Statistical analyses

All groups of discrete, single-measurement data were tested by Student's *t* test using GraphPad PRISM 4.03 (GraphPad Software Inc., La Jolla, CA). Insulin tolerance testing results were assessed by analysis of variance (ANOVA) with repeated measures using SPSS 17.0 (SPSS Inc., Chicago, IL). Overall survival was tested by logrank test, using GraphPad PRISM 4.03. Maximal survivorship was evaluated as previously described (*Wang et al., 2004*).

## Antibodies

The following antibodies were obtained for immunoblotting: p38 MAPK, phospho-p38 MAPK (Thr180/Tyr182), ERK, phospho-ERK (Thr202/Tyr204), JNK, phospho-JNK (Thr183/Tyr185), phospho-Akt (Ser473) and Akt, from Cell Signaling Technology (Beverly, MA); Nrf2 antibody from Novus Biologicals (Littleton, CO); β-actin, inhibitor from Sigma-Aldrich Corp.; and goat anti-rabbit and goat anti-mouse antibodies from Santa Cruz Biotechnology, Inc. (Santa Cruz, CA).

## Transcription factor analysis

Methods used to assess enrichment of transcription factor binding sites in 2 kb regions upstream of differentially expressed genes have been described in a recent publication (*Swindell et al., 2013*). In brief, we first assembled a dictionary of 1209 motifs representing the empirically-determined recognition sites of DNA-binding proteins (e.g., by chip-chip, chip-RNAseq, or protein binding microarray). These motifs were obtained by initially pooling those available in three databases, including Jaspar (145 motifs), UniPROBE (295 motifs) and TRANSFAC (819 motifs). This yielded an initial set of 1259 motifs, which were then filtered to exclude repetitive motifs or any motifs fewer than four base pairs in length, yielding the final set of 1209 motifs (*Swindell et al., 2013*). For each mouse gene, we retrieved the 2 kb upstream region based upon coordinates provided in UCSC ref gene files and sequences available from Bioconductor (package: BSgenome. Mmusculus.UCSC. mm10). Upstream sequences for all mouse genes were scanned for matches to each of the 1209 motifs, respectively (with masking of protein-coding sequences and repetitive DNA elements). Motif matches were identified using the 'matchPWM' function (R package: Biostrings), with an 80% match threshold, that is, a motif match was counted if the position weight matrix (PWM) similarity score exceeded 80% of the maximum score for that PWM. For each mouse gene, this yielded counts for each PWM indicating the frequency of motif matches in 2 kb upstream regions. The same procedure was applied to determine if motifs were enriched with respect to the 141 GHRH-KO-increased genes and the 164 GHRH-KO-decreased genes, respectively. To assess whether GHRH-KO-increased genes were enriched with respect to the number of occurrences for a given motif, for example, we used semi parametric generalized additive logistic regression models (*Swindell et al., 2013*). For these models, the response variable was an indicator with value 1 if a gene was included among the 141 GHRH-KO-increased, and with value 0 if the gene was not included among the 141 GHRH-KO-increased. Only liver-expressed genes were included in the analysis. Models included two predictor variables, including the number of motif occurrences in the 2 kb upstream region ($x_1$) and the length of sequence scanned ($x_2$). The variable $x_1$ was estimated using parametric methods, while $x_2$ was included as a non-parametric term with cubic spline smoothing. To assess motif enrichment, we evaluated the significance of the Z statistic associated with the coefficient estimate obtained for $x_1$. Models of this structure were generated for each of the 1209 motifs. To control the false discovery rate with respect to the 1209 tests, p values associated with Z statistics were adjusted using the Benjamini–Hochberg method.

## Indirect calorimetry

Mice were subjected to indirect calorimetry (PhysioScan Metabolic System from AccuScan Instruments, Columbus, OH) as described before (*Westbrook et al., 2009*). This system uses zirconia, infrared sensors and light beams arrays to monitor oxygen (VO2), carbon dioxide (VCO2), and spontaneous locomotor activity, respectively inside respiratory chambers in which individual mice were tested. All comparisons are based on animals studied simultaneously in eight different chambers connected to the same O2, CO2 and light beam sensors in an effort to minimize the effect of environmental variations and calibration on data. After a 24-hr acclimation period, mice were monitored in the metabolic chambers for 24 hr with ad libitum access to standard chow (Laboratory Diet 5001) and water. Gas samples were collected and analyzed every 5 min per animal, and the data were averaged for each hour.

## Assessment of blood chemistry

Plasma was obtained from blood collected by cardiac puncture at sacrifice and used for measurement of insulin using Mouse Insulin ELISA Kits (Crystal Chem, Downers Grove, IL). Following the manufacturer's protocol, total ketone bodies and non-esterified free fatty acids (NEFA) were measured using colorimetric assays from Wako Chemicals (Richmond, VA); glycerol was measured using kits from Sigma and triglycerides using kits from Pointe Scientific (Canton, MI), respectively. Adiponectin and resistin levels were assayed using Mouse Adiponectin/Resistin ELISA Kits (Linco Research, St. Charles, MO). Leptin levels were evaluated using Mouse Leptin ELISA Kits (Crystal Chem Inc., Downers Grove, IL). TNF-α and IL-6 were measured using Mouse TNF-α/IL-6 ELISA Kits (Biosource, Camarillo, CA). Plasma FFAs were assayed using optimized enzymatic colorimetric assays (Roche, Penzberg, Germany). Blood was taken from the tail to measure blood glucose using a glucometer (AgaMatrix, Salem, NH).

## Cytochrome P450 enzymatic activity assay

CYP1A and CYP2B activity was measured by resorufin conversion from methoxyresorufin (7 methoxy-resorufin O deethylation, MROD), ethoxyresorufin (7 ethoxyresorufin O deethylation, EROD), or pentoxyresorufin (7 pentoxy-resorufin O deethylation, PROD), as previously described (*Anwar-Mohamed et al., 2011*). Liver samples were collected from mice fed chow containing 1% tBHQ or control chow for 50 days that were exposed to 50 mg/kg diquat or saline for 6 hr prior to dissection. Unfrozen liver samples were briefly homogenized with a Potter-Elvehjem homogenizer and suspended in sucrose buffer. Microsomes were isolated via ultracentrifugation with the commercially available Endoplasmic Reticulum Isolation Kit (ER0100; Sigma-Aldrich). Changes in resorufin fluorescence were measured in a 96-well microtiter plates following the protocol outlined in the commercially available Cytochrome P450 2B Fluorescent Detection Kit (CYTO2B; Sigma).

# Additional information

### Funding

| Funder | Grant reference number | Author |
| --- | --- | --- |
| SIU | SIU CADRD Fund | Liou Y Sun |
| National Institutes of Health | AG031736, AG019899, AG038850 | Andrzej Bartke |

The funders had no role in study design, data collection and interpretation, or the decision to submit the work for publication.

### Author contributions

LYS, Conception and design, Acquisition of data, Analysis and interpretation of data, Drafting or revising the article, Contributed unpublished essential data or reagents; AS, YF, CH, Conception and design, Acquisition of data, Analysis and interpretation of data; WRS, Acquisition of data, Analysis and interpretation of data, Drafting or revising the article, Contributed unpublished essential data or reagents; JAH, JDB, RW, Acquisition of data, Analysis and interpretation of data; RS, Analysis and interpretation of data, Drafting or revising the article, Contributed unpublished essential data or

reagents; AB, Conception and design, Analysis and interpretation of data, Drafting or revising the article, Contributed unpublished essential data or reagents

### Ethics

Animal experimentation: This study was performed in strict accordance with the recommendations in the Guide for the Care and Use of Laboratory Animals of the National Institutes of Health. All of the animals were handled according to approved Institutional Animal Care and Use Committee (IACUC) protocols (#178-03-024) of the SIU School of Medicine.

## Additional files

### Supplementary files

• Supplementary file 1. (**A**) The table lists the 141 genes for which expression was most strongly elevated in Ghrh-KO mice as compared to littermate controls (FDR < 0.05 and FC > 1.50). The final column lists FDR-adjusted p values obtained by adjusting raw p values using the Benjamini-Hochberg method. (**B**) The table lists the 164 genes for which expression was most strongly decreased in Ghrh-KO mice as compared to littermate controls (FDR < 0.05 and FC < 0.67). The final column lists FDR-adjusted p values obtained by adjusting raw p values using the Benjamini-Hochberg method. (**C**) Primers used for qPCR analysis: phase I genes. Some of these sequences were previously published (*Sun et al., 2005*; *Sun et al., 2011*; *Steinbaugh et al., 2012*).

### Major dataset

The following dataset was generated:

| Author(s) | Year | Dataset title | Dataset ID and/or URL | Database, license, and accessibility information |
|---|---|---|---|---|
| Sun L, Swindell WR | 2013 | GHRH-KO liver | GSE51108; http://www.ncbi.nlm.nih.gov/geo/query/acc.cgi?acc=GSE51108 | Publicly available at GEO (http://www.ncbi.nlm.nih.gov/geo/). |

The following previously published datasets were used:

| Author(s) | Year | Dataset title | Dataset ID and/or URL | Database, license, and accessibility information |
|---|---|---|---|---|
| Amador-Noguez D, Darlington G, Yagi K, Venable K | 2004 | Transcription profiling of prop-1 and Ghrhr mutations in gene expression during normal aging in mice (Ames dwarf and Little mice) | E-MEXP-153; http://www.ebi.ac.uk/arrayexpress/experiments/E-MEXP-153/ | Publicly available at Array Express (http://www.ebi.ac.uk/arrayexpress/). |
| DeFord JH, Guigneaux MM, Boylston WH, Papaconstantinou J | 2005 | Aging Liver Profiles of Long Lived Pit-1 Mutant Mouse (Snell) and Wild Type Controls | GSE3129; http://www.ncbi.nlm.nih.gov/geo/query/acc.cgi?acc=GSE3129 | Publicly available at GEO (http://www.ncbi.nlm.nih.gov/geo/). |
| Dhahbi JM, Mote PL, Spindler SR | 2006 | Microarray profiling of potential calorie restriction mimetics | GSE2431; http://www.ncbi.nlm.nih.gov/geo/query/acc.cgi?acc=GSE2431 | Publicly available at GEO (http://www.ncbi.nlm.nih.gov/geo/). |
| Baur JA, Pearson KJ, Price NL, Jamieson HA, Lerin C, Kalra A, Prabhu VV, Allard JS, Lopez-Lluch G, Lewis K, Pistell PJ, Poosala S, Becker KG, Boss O, Gwinn D, Wang M, Ramaswamy S, Fishbein KW, Spencer RG, Lakatta EG, Le Couteur D, Shaw RJ, Navas P, Puigserver P, Ingram DK, de Cabo R, Sinclair DA | 2006 | Expression patterns in liver after resveratrol treatment of mice on a high-calorie diet | GSE6089; http://www.ncbi.nlm.nih.gov/geo/query/acc.cgi?acc=GSE6089 | Publicly available at GEO (http://www.ncbi.nlm.nih.gov/geo/). |

| | | | | |
|---|---|---|---|---|
| Yates MS, Tran Q, Dolan P, Osburn W, Shin S, McCulloch C, Silkworth J, Taguchi K, Yamamoto M, Williams C, Liby K, Sporn M, Sutter T, Kensler T | 2009 | Comparison of genetic and pharmacologic activation of Nrf2 signaling | GSE15633; http://www.ncbi.nlm.nih.gov/geo/query/acc.cgi?acc=GSE15633 | Publicly available at GEO (http://www.ncbi.nlm.nih.gov/geo/). |
| Yu RT, Zhang Y, Kliewer SA | 2012 | The role of FGF21 in aging and aging-related diseases | GSE39313; http://www.ncbi.nlm.nih.gov/geo/query/acc.cgi?acc=GSE39313 | Publicly available at GEO (http://www.ncbi.nlm.nih.gov/geo/). |

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
