## [Decision Letter]

Thank you for sending your work entitled “Growth hormone-releasing hormone disruption extends lifespan and regulates response to caloric restriction in mice” for consideration at *eLife*. Your article has been favorably evaluated by a Senior editor and 3 reviewers, one of whom is a member of our Board of Reviewing Editors.

The Reviewing editor and the other reviewers discussed their comments before we reached this decision, and the Reviewing editor has assembled the following comments to help you prepare a revised submission.

Sun et al. investigated the role of GHRH on lifespan and studied the endocrinological and cellular changes that may be responsible for the effect of GHRH on longevity extension. They show a major lifespan extension in GHRH-KO mice, comparable to that of GHR-KO mice. Also, they show that GHRH-KO mice are insulin sensitive. Among the possible protective mechanism presented for GHRH-KO is the decreased activity of S6K and the increased expression of xenobiotic genes. This is a very interesting paper that serves as a major and important extension of what has been shown for GHR-KO and GH deficiency. These data provide conclusive evidence that the GHRH-GH-GHR axis is the most potent pro-aging pathway discovered in mammals.

Major comments:

1) The lifespan of the wild-type mice is relatively short. The authors should discuss the reason for the short lifespan, comparing it to their previous publications, and should also emphasize/discuss the mixed strain background as a possible complicating factor in the analysis of these mice.

2) If phosphorylation of targets of S6K is used to test the hypothesis that its inactivation may cause insulin sensitivity in GHRH-KO mice, then this activity should also be measured in either muscle or adipose cells.

3) The authors fail to discuss causes of death – are the causes the same for the KO and wild-type mice? Is cancer incidence affected?

4) In Figure 6, the weigh curves for the male wild-type CR mice show that the male CR mice weighed almost the same as the AL mice. What is the explanation for this?

5) Given the dramatic changes in expression of the xenobiotic genes, it seems important to perform western blot analysis on at least a subset to determine to what extent the protein levels increase. Also, the tissue specificity is striking. It should be tested whether these changes are specific to liver, or whether they are apparent in other tissues, particularly metabolic tissues like skeletal muscle and adipose. In addition, it is not clear why the genes involved in xenobiotic metabolism are so important in mice that are not exposed to any xenobiotics. Is there any indication from the literature for why these genes may be important in the absence of toxic environments? This should be discussed.

6) The data in Figure 2 suggest that there is at least a trend toward higher oxygen consumption in the KO mice. While the data as generated may not reach significance, the authors should consider whether it really reflects no change, particularly in the daytime measurements.

---

## [Author Response]

*1) The lifespan of the wild-type mice is relatively short. The authors should discuss the reason for the short lifespan, comparing it to their previous publications, and should also emphasize/discuss the mixed strain background as a possible complicating factor in the analysis of these mice*.

The reviewers raise an important point. The remarkable magnitude of the longevity extension in GHRH-KO mice may be related in some ways to the relatively short lifespan of normal animals from this strain, approximately 21 months, as compared to approximately 24 months in normal siblings of Ames dwarfs and approximately 29 months in normal siblings on GHR^-/-^ mice in our colonies (10; 7; 8). Studies in GHR^-/-^ mice from Dr. Kopchick’s laboratory and in our laboratory indicate that the presence of a significant extension of lifespan in animals lacking GH signals does not depend on genetic background (three different backgrounds were tested, two mixed and one inbred). However, the magnitude of this effect tends to be reduced in a long-lived strain (11). Differences in the genetic background of GHRH-KO mice and Ames dwarf, Snell dwarf, GHR^-/-^, and little mice used in previous studies and different laboratories make it difficult to determine whether absence of GHRH, GH or their receptors have quantitatively similar or differential impacts on mouse lifespan. These comparisons are further complicated by differences between the diets used in different laboratories. These considerations are now discussed in the revised manuscript.

Moreover, mice used in these studies are maintained on a heterogeneous genetic background by avoiding brother x sister matings and genetic “bottle-necks” and thus producing animals that are heterozygous at a significant proportion of the loci (Panici et al., 2009). We feel that this more closely resembles the genetic constitution of animals in nature and also human populations, and thus we believe that our findings are more generalizable (or “translatable”) than data obtained in a single inbred strain. A similar strategy is well employed in the National Institute on Aging's Interventions Testing Program (ITP) (Harrison et al., 2009; Miller et al., 2007).

*2) If phosphorylation of targets of S6K is used to test the hypothesis that its inactivation may cause insulin sensitivity in GHRH-KO mice, then this activity should also be measured in either muscle or adipose cells*.

We thank the reviewer for this suggestion. We have performed the suggested experiments to compare the phosphorylation status of ribosomal protein S6 in skeletal muscle and white adipose tissue in GHRH-KO and control mice. The new data (now in Figure 2—figure supplement 1) have shown that adipose S6 phosphorylation is significantly reduced in GHRH-KO mice compared to controls, whereas there is no significant difference in muscle S6 phosphorylation between the two genotypes. These results imply that the inactivation of S6K in GHRH-KO mice might be tissue-specific.

*3) The authors fail to discuss causes of death – are the causes the same for the KO and wild-type mice? Is cancer incidence affected*?

We certainly agree that future end-of-life pathology studies aimed at identifying causes of death in these animals would be of great interest. We have preserved carcasses of these animals for histopathological studies and our collaborator will analyze these materials. However, these pathology studies will require a great deal of time to complete and thus we will need to include these results in a future manuscript. Nevertheless, extrapolating from findings in other long-lived mutants with reduced somatotropic signaling, we suspect that reduced incidence and/or delayed onset of cancer may have been involved. This is now mentioned in the revised manuscript.

*4) In*
Figure 6*, the weigh curves for the male wild-type CR mice show that the male CR mice weighed almost the same as the AL mice. What is the explanation for this*?

The expected pronounced difference of body weight between N-AL and N-CR males was seen up to the age of approximately 80 weeks, which approaches their median life expectancy. Afterward, body weight of N-AL animals began to exhibit a decline, which is normally seen during the final phase of life. The average body weights at advanced age may also have been influenced by deaths of some of the animals, especially in the N-AL group. The age-related decline of body weight in the N-CR group was delayed and diminished, as would be expected from the impact of CR on aging and longevity. The net result was that after the age of approximately 80 weeks, body weights of N-AL and N-CR males began to gradually converge. This is now stated in the manuscript.

*5) Given the dramatic changes in expression of the xenobiotic genes, it seems important to perform western blot analysis on at least a subset to determine to what extent the protein levels increase. Also, the tissue specificity is striking. It should be tested whether these changes are specific to liver, or whether they are apparent in other tissues, particularly metabolic tissues like skeletal muscle and adipose. In addition, it is not clear why the genes involved in xenobiotic metabolism are so important in mice that are not exposed to any xenobiotics. Is there any indication from the literature for why these genes may be important in the absence of toxic environments? This should be discussed*.

We appreciate the insightful comment of the reviewers. Regrettably, only a few of the xenobiotic enzymes can be measured by Western blot because specific antibodies are not currently available. We did, however, find an antibody specific for Cyp2b10, whose mRNA was seen to be elevated (Figure 4) in GHRH-KO mice, and used this antibody to measure Cyp2b10 expression in liver of GHRH-KO and littermate control mice by Western blot. We found that Cyp2b10 expression was significantly elevated in GHRH-KO mice relative to the controls also at the protein level (Figure 4—figure supplement 1) similar to the observed elevation seen at the mRNA level. In parallel, we also performed Resorufin conversion assay to measure cytochromeP450 enzymatic activity in liver tissues. GHRH-KO mice were found to have significantly higher hepatic cytochrome p450 activity than control mice (Figure 4—figure supplement 2).

We agree with the reviewers that these changes in xenobiotic metabolism enzymes are tissue-specific. These enzymes are usually more concentrated in the organs that are most directly exposed to the environment and toxins (i.e., liver, intestines, or lungs). We have found that other metabolic tissues including muscle and adipose have very low (undetectable) expression and activity of these xenobiotic enzymes in both GHRH-KO and control mice.

Lastly, the activation of xenobiotic metabolism pathways has been recently linked to lifespan extension in different slow-aging models ([3]; [24]; Shore and Ruvkun, 2013; [39]). However, the mechanisms underlying activation of xenobiotic metabolism genes in the absence of environmental toxins remain unknown. Some studies have shown elevation of bile acids (as potential endogenous xenobiotics) level and altered bile acid metabolism in GH deficient mice (3; 2). Moreover, several GH-mediated transcriptional factors, including Nrf-2, STAT 5b, and hepatic nuclear factor (HNF) 4alpha, have been shown to play an important role in xenobiotic gene expressions (53; 56). Thus, we postulate that activation of xenobiotic metabolism pathways, presumably through inhibition of GH signaling and alterations of the corresponding transcriptional factors, protects the tissues from the damaging effects of endogenous and exogenous molecules and thus contributes to maintaining tissue and metabolic homeostasis during aging. We have added this information to the Discussion section of our paper.

*6) The data in*
Figure 2
*suggest that there is at least a trend toward higher oxygen consumption in the KO mice. While the data as generated may not reach significance, the authors should consider whether it really reflects no change, particularly in the daytime measurements*.

We agree with the reviewers that KO mice seem likely to have higher oxygen consumption rate. However, further analysis did not reveal any statistical significance. We speculate that GH signaling-related alternations in energy expenditure (i.e., oxygen consumption, thermogenesis, heat loss) could be differentially regulated at various developmental stages.